# Lay perspectives of the open-label placebo rationale: a qualitative study of participants in an experimental trial

Cosima Locher ![ORCID] ,[1,2,3] Sarah Buergler,[2] Antje Frey Nascimento ![ORCID] ,[2] Linda Kost,[2] Charlotte Blease,[4] Jens Gaab[2]

[1]Department of Consultation-Liaison Psychiatry and Psychosomatics, University Hospital Zurich, Zurich, Switzerland
[2]Division of Clinical Psychology and Psychotherapy, Faculty of Psychology, University of Basel, Basel, Switzerland
[3]Faculty of Health, University of Plymouth, Plymouth, UK
[4]General Medicine and Primary Care, Harvard Medical School, Boston, Massachusetts, USA

**Correspondence to**
Dr Cosima Locher;
cosimaantoinette.locher@uzh.ch

## ABSTRACT

**Objectives** To analyse participants' concepts about the open-label placebo (OLP) effect; to explore their views about the discussion points that are applied in conventional OLP trials and to examine their experiences of taking part in an OLP trial.

**Design** A qualitative study using thematic analysis of semistructured interviews that were nested within a randomised controlled trial investigating experimental OLP analgesia (registered at ClinicalTrials.gov: NCT02578420).

**Participants** 30 healthy adults who took part in the randomised controlled trial.

**Results** Participants mostly conceptualised placebo as something that is inert and requires deception in order to be effective. Interviewees used a broad definition of placebos, going beyond a conventional notion of sugar pills. In contrast to the conventional OLP rationale, participants seldom emphasised classical conditioning as a mechanism of placebo effects, stressing a variety of other well-established components through which placebos might be therapeutic, whereas the conventional OLP disclosures state that 'a positive attitude helps but is not necessary', participants in our study applied other attitudes, such as 'it's worth a try'. When asked about their experiences during the trial, the majority emphasised that the concept of OLP was completely novel to them. Participants were rather sceptical about the efficacy of the intervention.

**Conclusion** Integrating lay perspectives into the scientific rationale of OLP treatments might enhance the plausibility and credibility of the rationale in ethical treatments.

**Trial registration number** NCT02578420.

## Strengths and limitations of this study

► Despite the increasing number of open-label placebo (OLP) trials, this is, to the best of our knowledge, the first qualitative study that examined participants' attitudes about OLP.
► The semistructured interviews generated rich, contextual data.
► The sample consisted of young university students, which may limit generalisability.

## INTRODUCTION

There is evidence that placebos exhibit substantial effects with promising clinical potential in various mental and physical reports.[1] Placebos in randomised controlled trials (RCTs) usually refer to dummy medications (eg, sugar pills) which in theory are indistinguishable from the treatment under investigation and which serve as methodological tools to screen out confounders that are associated with clinical trials (eg, spontaneous remission, regression to the mean).[2] In contrast, placebos in clinical practice are interventions without known specific effects on the treated condition but used with the goal of achieving positive outcomes by eliciting placebo effects (eg, through treatment expectations).[3] However, the potential of placebo effects in clinical practice is impeded as its administration requires deception. Some health researchers have argued that deceptive placebos (DPs) may sometimes be ethical in the interests of therapeutic gain—for example, where no other treatment options are available, or on the grounds that the deception is trivial and does not involve a major threat to patient autonomy.[4 5] However, most ethicists argue that deception in clinical context is unjustified and violates duties to be honest with patients.[6 7] In addition, deception may risk derailing trust in clinicians, leading to possible harms including disengagement with healthcare.[8 9]

Here, openly prescribed placebos offer the possibility to harness placebo effects in a transparent and, thus, ethical way.[10] The open-label placebo (OLP) approach has been examined in various physical and mental disorders, for example, chronic low back pain,[11 12] irritable bowel syndrome,[13] allergic rhinitis,[14–16] attention deficit hyperactivity disorder[17 18] and fatigue among cancer survivors.[19 20] So far, two meta-analysis examining OLPs revealed that patients in an OLP condition exhibit significantly greater improvement than those in a control group.[21 22] There is evidence that

OLPs may have significant effects on subclinical conditions, for example, menopausal hot flushes[23] and test anxiety[24 25] as well as on well-being, for example, sleep quality[26] and wound healing.[27] OLP has also been investigated in experimental settings such as experimentally induced pain and allergic reactions.[28–31]

All of these studies clearly inform participants that the intervention consists of 'honestly described placebo pills',[32] applying a plausible and positive rationale. A standard approach has emerged with these studies, whereby disclosure about the rationale conveys four discussion points, which are based on both study findings and clinical considerations[13 32]: (1) *placebos are powerful*. The major aim of this statement is to remove the stigma of placebo effects and to openly inform participants about the wide array of study findings that report powerful placebo effects in double-blind RCTs.[32] (2) *The body can automatically respond to taking placebo pills like Pavlov's dogs who salivated when they heard a bell*. The second discussion point emphasises classical conditioning as a major mechanism of how placebos might elicit therapeutic effects.[33 34] (3) *A positive attitude helps but is not necessary*. The aim of the third key point may be designed to take participants' scepticism seriously and to emphasise that they do not have to believe in the treatment for it to work.[32] (4) *Taking the pills faithfully is critical*. The final discussion point is related to study findings from RCTs that reveal that adherence is crucial for a (medical) treatment to work.[35 36] In this paper, we describe OLP studies that focus on these discussion points as 'conventional OLP' disclosures.

From an ethical point of view, transparency in the OLP rationale is crucial in order to respect participants' autonomy.[8 37] So far, two qualitative investigations have examined what participants experience when they are informed that they were assigned to the placebo arm in a placebo-controlled clinical trial.[38 39] These studies reveal that participants hesitate and doubt whether the placebo treatment has a potential to heal, yet many patients also spontaneously express hope.[38] Although informative, these findings are not transferable to the field of OLP, where participants know that they will receive a placebo. There is only one pilot study that examined participants' perspectives on the OLP treatment.[40] In that study, 10 participants were asked about their definition of placebo, whether the OLP rationale was clear to them, and whether anything about the study was misleading. None of the questions, however, solicited feedback on attitudes about OLP.

Therefore, this qualitative study set out to investigate participants' attitudes about (open-label) placebo. The study was nested within an RCT that investigated OLP analgesia in healthy participants. The aims of the study were three-fold: first, to analyse participants' concepts about the (open-label) placebo effect. Second, to compare participants' views on OLP with discussion points that are applied in conventional OLP trials. Third, to summarise participants' experiences of taking part in an OLP trial. For all our aims, we strove to compare participants'

statements of the three intervention groups of the quantitative study, that is, OPR without rationale (OPR−), OLP with rationale (OPR+) and DP groups.

## METHODS
### Study design
This is a mixed-methods qualitative study embedded within an RCT investigating the OLP treatment in healthy participants.[30] The study applied an experimental heat pain paradigm, comparing four groups, that is, no treatment, OPR−, OPR+, and DP. The rationale that was provided to the three groups can be found in the supplement (see online supplemental appendix 1). In the quantitative part that has been published elsewhere,[30] we found that the OPR + group exhibited a significant reduction of subjective heat pain ratings that did not differ from the DP group. For the qualitative part of the study, semi-structured interviews were scheduled at the end of each participant's treatment.

### Study participants
Thirty participants from the intervention groups (ie, OPR−, OPR+ and DP) out of the total sample of 160 were randomly selected to take part in the nested qualitative study. Participants were healthy adults from the general population recruited through advertisements. None of the participants was Psychology or Medicine students due to potential prior knowledge regarding placebo mechanisms and effects. Interviewing participants from the different intervention groups allowed us to examine attitudes towards the OLP approach in dependence on previous experimental experiences. Table 1 shows participants' basic demographic information. Recruitment started in January 2016 and lasted until July 2016. Written informed consent was obtained from each participant.

### Interview procedure
The interviews were conducted by a trained interviewer (CL) applying a semistandardised interview guide. All participants have been interviewed about their attitudes towards the (open-label) placebo effect. In order to ensure that participants who did not receive the OLP treatment (ie, OPR− and DP groups) are not primed by the existing OLP rationale, we aimed to explore their attitudes towards the placebo effect with a bottom-up approach. The interview covered the following components: (a) identity of the placebo effect, (b) possible causes of the placebo effect, (c) opinions about short-term and long-term consequences of the placebo effect, (d) the perceived control over the placebo effect and (e) the timeline of the placebo effect. A broad interview guide was prepared comprising 21 questions as a tool to obtain information (see online supplemental appendix 2). Interviews lasted between 21 min and 66 min. The interviews were audio-taped and verbatim transcripts were written. The listed quotations have been translated from German to English.

**Table 1** Sociodemographic characteristics of study participants

| Group | N | Age (SD) | N (%) female | Family status | Highest educational level N | Employment level N | Credibility of the intervention N |
|---|---|---|---|---|---|---|---|
| OPR– | 10 | 25.3 (6.63) | 6 (60%) | Single: 10 | Primary school: 0<br>Secondary school: 1<br>High school: 5<br>University: 4 | Full time: 0<br>Part time: 3<br>None or student: 7 | Credible: 8<br>Somewhat credible: 2<br>Not credible: 0 |
| DP | 10 | 30.9 (14.65) | 6 (60%) | Single: 9<br>Divorced: 1 | Primary school: 1<br>Secondary school: 0<br>High school: 5<br>University: 4 | Full time: 0<br>Part time: 5<br>None or student: 5 | Credible: 6<br>Somewhat credible: 3<br>Not credible: 1 |
| OPR+ | 10 | 25.4 (4.86) | 7 (70%) | Single: 10 | Primary school: 0<br>Secondary school: 1<br>High school: 5<br>University: 4 | Full time: 1<br>Part time: 5<br>None or student: 4 | Credible: 9<br>Somewhat credible: 1<br>Not credible: 0 |

DP, deceptive placebo; OPR+, open-label placebo with rationale; OPR–, open-label placebo without rationale.

The first author who conducted the interviews (CL) completed her PhD in 2017. In her research, she focuses on the effects, mechanisms and routes of administration of (open-label) placebo. For that, she applies different methodological approaches, encompassing qualitative studies, experimental trials, clinical investigations and (network) meta-analyses.

### Qualitative analysis

For data analysis, MAXQDA, VERBI Software, version 2019 was used.[41] We followed an inductive-deductive hybrid approach[42] and incorporated primarily an inductive data-driven approach to allow the examination of core themes for a phenomenon with limited existing theory or research literature,[43–45] complemented by a deductive ideation of the theoretical points of the OLP treatment rationale.[46] With the aim to describe common patterns across the interviews, the following six steps were performed[47]: first, the comment transcripts were read 3–4 times by four coders (SB, LK, AFN and CL) in order to achieve familiarisation with the data. Second, the two main coders (SB and LK) identified brief descriptive labels ('codes') that represented different aspects of interviewees' descriptions. More than a single code was applied if quotations had multiple meanings. Third, the four coders (SB, LK, AFN and CL) examined the codes and sorted them into different themes with shared meaning units. Fourth, the themes were reviewed, that is, themes that were not supported by the data were modified by the coders. Fifth, the coders developed a detailed analysis of each theme by defining final names. Finally, all authors took part in authoring the manuscript. We sought analytical rigour by involving multiple coders.

### RESULTS
### Participants

Participants had a mean age of 27.2 (SD 9.73) years, and 63% of the participants were woman (see table 1). The three groups were comparable regarding age (F(2,27) = 1.09, p=0.35), sex ($X^2(2)=0.29$, p=0.866), family status ($X^2(4)=3.61$, p=0.462), educational level ($X^2(6)=3.00$, p=0.809) and employment level ($X^2(4)=3.49$, p=0.479). Quantitative results were reported elsewhere.[30 48] The majority of participants assigned to the OPR– and OPR + groups believed they were receiving a placebo cream (17/20). Most participants in the DP group also believed they were receiving the analgesic cream (6/10).

### Overview

In the 30 interviews, 667 text passages of 30'379 words were identified and summarised into 65 categories. These categories were subsumed in eight main categories, which were classified in three superordinate categories: (1) the placebo concept, (2) OLP rationale and (3) experiences of taking part in an OLP trial (see table 2). An overview of all superordinate categories, main categories and categories is provided in the supplement, also revealing the number of quotations for each category per group (online supplemental appendix table 1). In the

**Table 2** Superordinate categories with corresponding main categories

| Superordinate categories | Main categories |
|---|---|
| 1. The placebo concept. | 1.1. Descriptive |
| | 1.2. Effectiveness |
| | 1.3 Specific examples |
| 2. Open-label placebo rationale | 2.1 First discussion point: 'Placebos are powerful' |
| | 2.2 Second discussion point: 'The body automatically responds to placebos like Pavlov's dog' |
| | 2.3 Third discussion point: 'A positive attitude helps but is not necessary' |
| 3. Experiences of taking part in an open-label placebo trial | 3.1. Reaction to (open-label) placebo |
| | 3.2. Efficacy and effectiveness during the experiment |

following, we present the four superordinate categories and main categories. For the complete list of categories, please see table 2.

## Placebo concept
### Descriptive
The vast majority of participants agreed on the definition of placebo as something that is characterised by its inertness and its deceptive administration. This assumption was independent of group allocation; for example:

> To me, a placebo is a drug without actual ingredients (DP; participant 42)

Likewise, many participants emphasised that deception is a prerequisite for placebos to work. Notably, examples not only included the classical placebo pill but also deceptive aspects in politics and advertisements that lead to a change of belief or behaviour; for example:

> Well, in term of deception - so when you use the term deception, I don't know, in politics that happens; I often feel like you can achieve something with deceptive means (OPR−; participant 89)

A few comments linked the placebo concept to the design and conductance of clinical studies, such as randomised controlled trials; for example:

> Well, I've heard that quite often now, that, I don't know, experiments or other studies were conducted where people were given placebos (OPR−; participant 56)

Notably, two participants from the OPR− group expressed the view that placebos are like a copy or a substitute of a verum intervention, which is, however, less powerful:

> Yes, some kind of substitute, like - just like this clone phenomenon, that is kind of like a copy but not quite the same, and also not as good <<laughs>> (OPR−; participant 135)

### Effectiveness
Interviewees expressed a variety of perspectives about the effectiveness of placebos. Especially participants who were allocated to the OPR+ and OPR− groups emphasised the paradox that placebos are inert, yet reveal positive effects; for example:

> When I think of a 'drug' it's a drug that doesn't have any medical ingredients; it's like neutral actually yet triggers an effect afterwards (OPR+; participant 141)

Similarly, many participants indicated general agreement that the intake of placebos has an effect on the body. This was independent of group allocation; for example:

> [Mhm] (.) more than an imagination actually, with actual physical effects (OPR+; participant 109)

In opposition to this view, other interviewees neglected that placebos can have a physical effect. Some interviewees stressed that placebos are genuinely free of effect, also using terms like 'sham' and 'fake'. Notably, except one participant, all statements stemmed from participants assigned to the OPR− or DP groups; for example:

> No no no, it's really only controlled by the mind in our head, so one should not be able to measure anything purely physically (OPR−; participant 89)

### Specific examples
Building on this theme, the vast majority of interviewees offered a variety of examples of, what they considered to be, placebos. Examples went beyond the classical pill encompassing autosuggestion, esotericism, complementary medicine, food supplements, advertisement, mindfulness and food. Notably, patients allocated to the OPR + group gave more context to these statements, explaining why they think that these kinds of placebos might work:

> Yes, it probably is the case that sometimes, when I notice that I'm getting a cold or have a cold, I just take and eat ginger, and I'm convinced that it works every single time. In this case I also feel like it works <<laughs>>; and I can certainly imagine that it just…that these rituals also have an effect, yes (OPR+; participant 157)

More broadly, and often related to daily life experiences, interviewees also mentioned that a placebo can be a general act, an activity, or something that happens in the interpersonal contact; for example:

> The sauna, for example, I do it regularly as soon as winter starts. And was never sick ever since (OPR−; participant 74)

> I always watch TV in order to fall asleep. I fall asleep faster when I do that than when I don't. A placebo is also like a ritual, yes (OPR+; participant 146)

## OLP rationale
All interviewees stressed components that can be linked to the established OLP rationale, whereas some aspects were in line with the discussion points of the rationale, others were complementary or even contradictory:

### First discussion point
The majority of interviewees concretised the anticipated effect of an OLP treatment, which is related to the first discussion point of the OLP rationale (ie, *placebos are powerful*). Participants agreed that an OLP treatment is less effective and/or has shorter lived effects than the DP treatment; for example:

> 'But I feel like if you know that it's a placebo, it may not be as long-lasting than if you don't' (OPR+; participant 146)

More precisely, interviewees stated that key mechanisms such as positive expectations and belief systems are

reduced in the OLP treatment. It was also argued that the lack of deception is associated with a significant reduction of placebo efficacy. Most concerns were expressed from participants assigned to the OPR− and DP group with the exception of two people from the OPR + group; for example:

[Hmm] well, I believe the whole belief system collapses a little bit; I mean when you have a very medical name and an appealing packaging, it will work even if there's a placebo pill inside (OPR−; participant 74)

For me a placebo is something that is really connected to a strong belief, so I have a feeling that it only works, or especially works, when for example one doesn't know that it is a placebo (OPR+; participant 141)

Some interviewees from the OPR + group, however, believed that an OLP treatment can be powerful; for example:

Yes, even if it's just a placebo, uhm, simply from the studies that have been conducted so far, one knows that it can work; and already that, that statement has had an effect on me and I thought 'well okay so even if it's open it still works a little bit' so (OPR+; participant 35)

### Second discussion point

Interviewees listed a big variety of mechanisms that make the OLP treatment work.

In line with the second discussion point of the established OLP rationale (ie, *the body can automatically respond to taking placebo pills like Pavlov's dogs who salivated when they heard a bell*), some interviewees emphasised classical conditioning as a key mechanism for placebos to be effective. Notably, only participants from the OPR + group explicitly referred to the term 'conditioning', whereas others talked about the impact of experiences and unconscious processes; for example:

Yes, because we're used to it, just like for example when I saw the drug and had the package in my hand, yes for me it's association so it's a drug, only drugs look like that (OPR+; participant 35)

Notably, however, comments that could be linked to 'conditioning' represented only a minority of interviewees' statements. The vast majority highlighted the impact of personality traits such as openness and a positive attitude. This was independent of group allocation; for example:

I think I'm pretty open-minded regarding new things, maybe that has something to do with it, too (OPR+; participant 141)

Besides people's personality components and general attitudes, interviewees also stressed personal mindsets that are directed towards the future, encompassing positive expectancies, beliefs and hope; for example:

[Hmm] or from the expectations one might have; I can imagine that simply through the doctor administering something, like a preparation, people will generate certain expectations (OPR+; participant 109)

More broadly, participants acknowledged the mind-body connection and psychological components in general; for example:

Well, it's not that something just happens by mind power, but something in my body changes through mind power. So, there's definitely something happening on a molecular basis, that's clear (DP; participant 154)

Besides components that are linked to the individual patient, other comments underlined the impact of a trust-based patient–provider relationship. Again, this was independent of group allocation; for example:

So, I took it when I was a kid because my doctor recommended it to me, and I mean as a kid you trust doctors 100% anyway, right?<<laughs>> and I've always felt very comfortable in that practice, that's also very important. And the investigator [of the study] was very nice and always explained exactly what she was doing, I think that there's a big connection (OPR−; participant 99)

Absolutely, so I had one [an investigator] who had long conversations with me, and consequently one is-you're more easily inclined to engage in something when you feel like you're in good hands and he really took his time too so, yeah (DP; participant 48)

Some interviewees highlighted other placebo mechanisms such as the power of imagination and the ritual. Notably, one participant also emphasised that a response to receiving a placebo might also encompass the natural course of the disease:

So, in a first step someone takes a placebo, and by pure chance their headache fades away. Well not in that exact moment, but after a couple of minutes, on its own, right? [interviewer: like a natural course?] Exactly. So, the pain comes and goes without taking any painkillers (OPR−; participant 39)

### Third discussion point

The third discussion point of the original OLP rationale (ie, *a positive attitude helps but is not necessary*) has the general aim to take participants' scepticism seriously and to operate as a kind of relativisation. Crucially, none of the participants spontaneously expressed such a thought. However, they referred to other aspects of relativisation that would encourage them to take an OLP, such as the idea of 'nothing to lose' and 'it's worth to give it a try'; for example:

I'd probably do it in a way where I would give the cream a chance and say, 'it doesn't do me any harm, just give it a try.' (OPR−; participant 89)

You actually can't lose anything at all, that's another aspect of it, I think. It's definitely worth a try (OPR+; participant 109)

Well, if it helps and doesn't harm my body in any way, then that's great (DP; participant 138)

Others agreed that the OLP treatment would be an option for them *if* there would be no alternative; for example:

Good question, I think it depends on the situation, like how alarming it is. If I wouldn't have anything but the placebo cream, I'd probably use it<<laughs>> (OPR−; participant 135)

On a broader scale, interviewees speculated that the readiness to take an OLP treatment also depends on the context and the respective disorders; for example:

As I said, you need to differentiate of course, as there are different forms of diseases; when I talk about the common cold or cancer therapy, those are clearly two different things, and of course you have to differentiate (DP; participant 138)

### Experiences of taking part in an OLP trial
#### Reaction to (open-label) placebo
The vast majority of participants assigned to the OPR + group emphasised that the OLP approach was completely novel to them; for example:

Um so I didn't know that there's research - well that there's research showing that there are effects when you know that it's a placebo and it still has an effect (OPR+; participant 146)

Furthermore, patients who were allocated to the OPR + group agreed that the OLP treatment was only plausible because of the rationale that was provided during the experiment; for example:

How should I say, that there's still the release of some substances; now I'm not very familiar with this <<laughs>>. But uhm I think when she [the investigator] told me about it, that it's like conditioned, that seemed plausible to me (OPR+; participant 141)

Opposing to this, a dominant theme in the OPR− group was the conviction that the OLP treatment is a little strange, crazy and incomprehensible; for example:

But that's not negative at all, if anything I think it's a funny farce <<laughs >>" (OPR-; participant 135)

"It is very peculiar (OPR−; participant 74)

Interestingly, three participants (ie, two from the OPR− group and one from the OPR + group) stated that they were speculating during the experiment whether they really received a placebo; for example:

You might also think 'ah maybe it's not a placebo, maybe it's real cream and they won't tell me just to

see- maybe to see how I react when I don't know that it contains active ingredients, and then has an effect anyway' (OPR−; participant 57)

#### Efficacy and effectiveness during the experiment
Finally, interviewees also made estimates about the efficacy of placebos in the experiment. The majority of participants of the OPR+ and OPR− groups suggested that the intervention was probably not effective. Some participants from the DP group shared this impression; for example:

No, I don't think so <<both laugh >>, I don't think it worked, I even checked: it [the skin] was red right away, so yeah, I don't think it really worked (OPR−; participant 57)

To be honest, I don't think it made a difference (OPR+; participant 34)

I did not notice a difference (DP; participant 75)

On the contrary, some participants of the DP group reported that the intervention had an effect on their pain levels; likewise, some interviewees from the OPR+ and OPR− groups suggested that the intervention had an effect on their mind; for example:

I really think, psychologically, because I um was told that with open placebos, well, the trials where people knew it's a placebo, that it worked there. I think it was through that statement actually, yes (OPR+; participant 35)

Some participants were also unsure about whether the intervention had an effect or not; for example:

Well, I was wondering if it had an effect on my pain threshold, I don't know the results. And um, I tried to convince myself that it [the placebo] doesn't have an effect, I should be on the same threshold level [in the second round], yet there was this little thought 'it could still be effective' (OPR−; participant 89)

Notably, some interviewees reported that experimental components such as the cooling of the cream or the patch position had an influence on their pain experience; for example:

But maybe I just thought um, just the cream, not the active ingredient is having an effect, just the layer <<laughs >> between the plate and the skin (OPR−; participant 89)

### DISCUSSION
This qualitative study set out to evaluate participants' concepts about the (open-label) placebo effect, to compare participants' attitudes towards (open-label) placebos with the three discussion points of the conventional OLP disclosures and to summarise participants' experiences during the experimental OLP trial. The

present study is one of the first qualitative studies that is nested within an OLP trial.

We found that the vast majority of participants describe placebos as an intervention that is medically inert with respect to its ingredients for treating symptoms but efficacious and powerful. This is also in line with a pilot study that examined 10 patients who received OLPs as an adjunctive treatment to opioids for acute pain. There, the majority of participants described placebos as something that is inactive.[40] To define a placebo as an inert substance that has no effect on human physiology is common, yet questionable. Several researchers argued that no substance is inert if 'inert' is defined as something that has no physiological effect.[49–51] Inertness may also be associated with something that is neutral or unspecific. Placebos, however, reveal specific effects and mechanisms and can thus be conceptualised as a specific agent.[52] These contrasting views raise the question of whether it would be helpful to inform participants in OLP trials that placebos are specific by their nature. Another component that has been traditionally linked with the placebo term and that has been raised by the participants of our study is the assumption that placebos need deception in order to work. Particularly in clinical practice, however, it is clearly unethical to deceive patients by prescribing placebos as deception violates the principle of respect for patient autonomy.[53] In contrast, our results reveal that the long-time association of placebo and deception may be common in the general population. This implies that the information given in OLP trials, that is, that placebos can also work when delivered openly, is a crucial component of the OLP treatment. Interviewees described several kinds of placebos, going beyond the classical sugar pill. Notably, the spectrum of examples encompassed several aspects that have been studied in placebo research. The power of autosuggestion, for example, has been explained by a change of mind set and a reappraisal of a situation.[54] Likewise, the practice of mindfulness has been supposed to modify mind sets and has shown to be related to positive health outcomes.[55] Interviewees also related supportive interpersonal aspects to placebo effects, which is in line with research indicating that the alliance between the treatment provider and treatment recipient is related to the effects of the respective treatment, even if this treatment is placebo.[56–59] Finally, as outlined by some participants, religiosity and spirituality are related to hope, positive thinking and meaning-making, all mechanisms that have been suggested to operate in placebos.[60] Taken together, interviewees spontaneously outlined several kinds of placebos that have been the subject of multiple research approaches; this is in line with recent attempts to generalise placebos and to study them outside the medical context.[8]

The statements given by the interviewees were further compared with the conventional OLP rationale. Interviewees did not explicitly engage with the first discussion point (ie, *placebos are powerful*) despite interviewer attempts in the OPR + group, that is, most made no

statements in relation to the effectiveness of placebos, yet agreed that *openly* prescribed placebos might be less effective than DPs. This is in contrast to some first indications showing that openly described placebos work as well as DPs.[30 61] In order to inform patients that honestly prescribed placebos are also powerful, newer OLP trials started to outline the findings from the first controlled OLP trial,[13] usually showing a video sequence of an American NBC news report of the respective study.[11] Our findings indicate that it is of utmost importance to inform participants about the promising evidence of OLP trials.

In relation to the second discussion point (ie, *the body can automatically respond to taking placebo pills like Pavlov's dogs*), which clearly refers to one of the most important mechanisms of placebo effects (ie, classical conditioning), quotations from our participants were surprising. We found that only one participant from the OPR + group mentioned conditioning. Evidence in the field of conditioning also suggests that behaviour cannot only be triggered by verbal information, yet also by non-conscious stimuli, a differentiation that was not mentioned by our participants. Besides classical conditioning, the second most dominant conceptual model to explain placebo effects is expectancy,[62 63] a mechanism that was frequently stressed by our interviewees, also in relation to general beliefs and habits. A third mechanism that has received considerable attention and has shown to induce placebo effects is the patient–provider relationship,[3] a component that was also emphasised by many of our interviewees, especially in relation to trust. Notably, participants of our study stressed a wide array of other mechanisms through which placebos work, including general attitudes, beliefs, hope, the ritual and the power of imagination. All these factors are established mechanisms in placebo research,[64] usually subsumed under external and internal elements of the treatment context.[65] Another psychological component that received less attention from placebo researchers and that was frequently addressed by our participants is hope. This is in line with other qualitative placebo (yet not openly prescribed placebo) studies reporting that patients spontaneously expressed hope, which they usually explicitly distinguish from expectation,[38 66 67] whereas expectancy is linked to cognitive states and a prediction of the future, hope is not so much a prediction but rather more an emotional state.[38] Taken together, our findings reveal that participants of an OLP trial intuitively list a wide array of established placebo mechanisms. This raises the question why classical conditioning is *exclusively* stressed by the conventional OLP disclosures. A recent survey study that examined participants' preferences of placebo explanations found that other mechanisms than classical conditioning (ie, positive expectations and brain mechanisms) were rated most preferable.[68] Mechanisms that have been shown to be relevant for the efficacy of placebos cannot be transferred to the field of openly prescribed placebos. It has been argued that patient expectations about the efficacy of a treatment substantially differ between conventional and

OLP approaches.[29] Thus, it might be worth considering informing participants in OLP trials about the variety of mechanisms that operate in *openly* prescribed placebos. This will ensure that the provided information is based on newest evidence.

Finally, the third discussion point of the conventional OLP rationale (ie, *a positive attitude helps but is not necessary*) aims to reduce scepticism and works as a kind of relativisation. Although interviewees of our study expressed different strategies to reinforce realistic (and not too high) expectations, none of them used the reasoning that *a positive attitude helps but is not necessary* or comparable relativisation strategies. Interviewees argued that 'it's worth a try'; 'if it helps and does no harm' or 'worth to see what happens'. These viewpoints are not new: patients in randomised, sham-controlled trials who do not know whether they are assigned to the verum or the sham condition often argue that it is worth giving it a try.[38 39 69] The underlying heuristic appears to be: 'I lose little if the placebo does not work, but gain a lot if it does work'.[70] Notably, some participants also outlined that they would take a placebo 'if it's the only option'. This is related to the notion that their openness would 'depend on the symptoms or disorder', a finding that is in line with a previous qualitative study, applying a scenario to explain OLP to participants.[71] Patients with chronic primary pain, for example, often describe that different treatment approaches have turned out to be unsatisfactory, leading to a perceived lack of alternatives and a state of demoralisation,[72] which might lead to an openness towards OLPs. Therefore, the idea of informing participants that *a positive attitude helps but is not necessary* might be problematic: not only do participants use other strategies to reduce scepticism, they are also convinced that positive attitudes are an important mechanism for placebos to work—a sentiment which is supported by current scientific evidence.[73]

Finally, we were interested in participants' experiences in an experimental OLP trial. We found that for the vast majority of interviewees, the OLP approach was completely novel. Most interviewees had the impression that the treatment was non-effective. Slightly more participants from the DP group stated that the efficacy of the intervention was possible or even likely. Especially participants assigned to the OPR + group mentioned that the efficacy of the intervention was 'unclear'. Interestingly, slightly more participants from the OPR– group stated that the efficacy of the intervention is 'possible' when compared with the OPR + group. These rather more sceptical and reserved expressions related to the efficacy of the intervention, that we found independent of group assignment, are in contrast to results of the original RCT. There, groups with a rationale (OPR + and DP) reported significantly lower heat pain intensity and unpleasantness ratings than the OPR– group.[30] Beyond the presumed efficacy of the intervention, there were some group-dependent reactions to the concept of OLP, whereas participants from the OPR + group stressed that

the plausible rationale was probably the most important component of the treatment, participants from the OPR– group said that the intervention was 'strange, crazy and incomprehensible'. The differing views of these two groups underscore the importance of a plausible explanation of the concept of OLPs to counter possible scepticism.

## Strengths and limitations

The present qualitative study has a number of limitations. The OLP trial recruited young university students, and this may limit generalisability to other demographic populations. The participants were well educated and informed about placebos, which was reflected in the sophisticated answers of the participants (eg, only one participant was not aware of the term 'placebo'). A recent survey study also found that a higher education is related to more placebo knowledge.[68] Second, the present qualitative study examined healthy participants with presumably lower motivation for pain relief than patients. Likewise, the study did not investigate attitudes of participants towards practitioners who offer OLPs, or about possible negative effects, such as feelings of stigmatisation. Some patient populations may be highly resistant to being offered OLPs or offended by the idea. Third, although we did set out to compare participants' views of OLPs with the conventional OLP disclosures, we were not able to make conclusions about the fourth discussion point (ie, *taking the pills faithfully is critical*) since the topic of adherence was not applicable in our experimental RCT. Fourth, the researcher conducting all interviews were not blind to the group allocation, probably leading to a bias in the framing of the questions. Therefore, we decided that the two main coders were neither researchers who delivered the treatment nor those who conducted the interviews. Despite these limitations, this qualitative study offers a novel perspective on participants' views about openly prescribed placebos and contributes to the discussion regarding the use and framing of the OLP treatment rationale. The qualitative approach that we chose to reflect on the conventional OLP disclosures is one possibility. Other ways to further make suggestions for the adaption of the rationale might be to integrate newest evidence-based findings of OLP trials or to focus on ethical considerations.

## CONCLUSION

Participants of our experimental OLP trial spontaneously expressed a wide array of placebo definitions, examples and underlying mechanisms. In their descriptions, they went beyond theoretical speculations and provided a variety of personal experiences. Given the big variety of conceptualisations, but upholding standards of evidence, clinical practice might profit from tailoring the provided information to the specific needs of individual patients, which is in line with the recommendations formulated by a placebo expert group.[74] We were

able to link participants' quotations to the conventional OLP rationale. Overall, we propose that the OLP rationale would gain from further adaptions and should be revisited. We suggest that participants of OLP trials should be informed of *multiple* placebo mechanisms that exist, including but not limited to classical conditioning. Conceivably, this might further increase the plausibility of the rationale and will also be in line with newest evidence. Furthermore, and in order to set realistic expectations, we propose that participants should be motivated to 'just give it a try'. This is in contrast to the current strategy of the rationale to decrease possible scepticism. Considering that this qualitative analysis stems from an experimental trial, clinical implications should be outlined with caution: our findings provide preliminary indications that the OLP approach might only be suitable for a subgroup of patients who are not too sceptical about the efficacy of the OLP therapy and who are ready 'to just give it a try'—maybe also because they feel that OLPs are their last treatment option. Further research should move beyond conceptualisations of placebos, and opinions about their effectiveness, to attitudes about the acceptability of being offered OLPs, especially for patients living with a variety of placebo-responsive symptoms. Such studies should explore perceptions about the competence of clinicians who offer OLPs. We also recommend that future research solicits the views of patients drawn from different demographic populations.

**Acknowledgements** The authors would like to thank Xenia Knüsel and Sebastian Hasler for their assistance with the data transcription. They would also like to thank Berfin Bakis for her help with the translation of the quotations. We also thank Mareike Rytz for her help with the references.

**Contributors** CL, CB and JG: study concept and design; CL: conducted all interviews and drafted the manuscript; CL, SB, AFN and LK: coded all interviews; CB, JG, SB, AFN and LK: contributed to critical advice and revisions of the manuscript. All authors read and approved the final manuscript.

**Funding** CL received funding from the Swiss National Science Foundation (SNSF): P4P4PS_194536.

**Competing interests** None declared.

**Patient and public involvement statement** Patients and the public were not involved in the design of the study; however, the study specifically focuses on participant perceptions.

**Patient consent for publication** Obtained.

**Ethics approval** The study was approved by the Local Ethics Committee of the Canton Basel, Switzerland (EKNZ 2015-246).

**Provenance and peer review** Not commissioned; externally peer reviewed.

**Data availability statement** Data are available in a public, open access repository. Anonymized transcripts of the qualitative interviews are stored at the *Harvard Dataverse* data repository: https://doi.org/10.7910/DVN/ET5PVQ. Anonymized transcripts of the qualitative interviews are stored at the Harvard Dataverse data repository: https://doi.org/10.7910/DVN/ET5PVQ.

**ORCID iDs**
Cosima Locher http://orcid.org/0000-0002-9660-0590
Antje Frey Nascimento http://orcid.org/0000-0001-7907-0439

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
