## [Reviewer comments · BMJ Open]

ARTICLE DETAILS

TITLE (PROVISIONAL)	Lay perspectives of the open-label placebo rationale: A qualitative study of participants in an experimental trial
AUTHORS	Locher, Cosima; Buergler, Sarah; Frey Nascimento, Antje; Kost, Linda; Blease, Charlotte; Gaab, Jens

VERSION 1 – REVIEW

REVIEWER	Adam Geraghty University of Southampton, Primary Care and Population Sciences
REVIEW RETURNED	23-Jun-2021

GENERAL COMMENTS	Overall: This is a generally well written and interesting paper, that will be of use to those working in the placebo field. The focus on open label placebos is timely. However, I think there are a number of points where more detail is needed. See below. Abstract: 1) The abstract starts with “To analyse participants’ concepts about the open label placebo”... However we don’t know who the participants are? Patients? Health care professionals? Students? This needs to be defined. Perhaps mentioned in the first line, but then fully defined in the participant section of the abstract. 2) The design section of the abstract should state the RCT and the aim of the RCT. E.g. Design should state qualitative study nested with an RCT etc. Introduction: 3) The introduction is nicely written and clear. However, I don’t think you can avoid a definition of placebo in the opening section. As authors will know, defining placebo is a complex issue, with a range of opinions. However, it is important to know how the authors are defining placebo in this context. I think they need to add their definition of placebo to the opening paragraph. Methods: 4) I know this is a nested study, but this qualitative study needs to stand alone re information needed by the reader. We do not want to have to have the trial design open to understand the present study fully. In the study participant section, please add how they were recruited to the trial, adverts? Were they students? Etc.
---

	Discussion: 5) In the discussion, authors state the below on line 463: “Whereas expectancy is linked to cognitive states and a prediction of the future, hope is not so much a prediction but rather more an existential stance [36]” It would be useful if they could go further and unpack what they mean by existential stance, do they mean a trait? Further explication here would be helpful for readers. 6) I think it would be useful to discuss the implications of this work further. Although implications are discussed in relation to the rationale provided for open label placebos, and this appears to be the key focus, I think the authors, albeit briefly need to look beyond that. Is the aim of this research to make open label placebos a part of clinical practice? If so, I think there should be some consideration given for what the findings of this study might mean for application in clinical practice. For instance, I think it is important that most seem to express scepticism.
--	---

REVIEWER	Zhen Zhou University of Tasmania Menzies Institute for Medical Research
REVIEW RETURNED	27-Jun-2021

GENERAL COMMENTS	Thank you for inviting me to review this work. This is a very nicely written paper with important information conveyed that will provide many implications for the improvement of OLP rationales. The study methods and findings were well presented and the authors comprehensively and meticulously discussed the study findings. I only have few minor suggestions for the authors. First, the authors may consider clearly stating the rationales that were given to the OPR+ group in the OLP trial where their interviewees had been recruited (the first three rationales mentioned in their introduction). This may help readers to understand how these informed rationales may impact the OPR+ group’s responses to the questions related to the discussion points. Please ignore this suggestion if the authors already did that. I wonder it may be worth mentioning that the OLP trial in which their study was nested had not included Psychology or Medicine students because they potentially have a previous knowledge of placebo mechanisms and effects? It would be useful if the authors can add the concrete number of participants after some terms they used such as ‘the vast majority of interviewees’, ‘The majority of?’ I know they had this information included the appendix, but it often takes time for readers to find these information in the supplementary documents. In the first paragraph of the discussion section, the authors stated that their study compared participants’ attitudes toward OLP with the four discussion points. Shouldn’t it be ‘three’ discussion points as they claimed that they were unable to explore the fourth point due to the experimental design of their OLP trial? Also in this paragraph, the authors mentioned the results of the OLP trial, which are not the findings of their qualitative study. They may consider moving this part to the method section after the brief introduction of the trial.
---

VERSION 1 – AUTHOR RESPONSE

Reviewer: 1 (Dr. Adam Geraghty, University of Southampton)

Overall:

- This is a generally well written and interesting paper, that will be of use to those working in the placebo field. The focus on open label placebos is timely. However, I think there are a number of points where more detail is needed. See below.
- o We thank Dr. Geraghty for the positive feedback.

Abstract:

- The abstract starts with “To analyse participants’ concepts about the open label placebo”... However we don’t know who the participants are? Patients? Health care professionals? Students? This needs to be defined. Perhaps mentioned in the first line, but then fully defined in the participant section of the abstract.
- o We agree with the reviewer that this information was missing in the abstract. The following sentence was added: “Thirty healthy adults who took part in the randomized controlled trial.”
- The design section of the abstract should state the RCT and the aim of the RCT. E.g. Design should state qualitative study nested with an RCT etc.
- o We agree with the reviewer and, therefore, described now in detail: “A qualitative study using thematic analysis of semi-structured interviews that was nested within a randomized controlled trial investigating experimental OLP analgesia.”

Introduction:

- The introduction is nicely written and clear. However, I don’t think you can avoid a definition of placebo in the opening section. As authors will know, defining placebo is a complex issue, with a range of opinions. However, it is important to know how the authors are defining placebo in this context. I think they need to add their definition of placebo to the opening paragraph.
- o We thank the reviewer for pointing this out. Therefore, we decided to include a detailed definition of placebo in the first paragraph of the introduction: “Placebos in randomized controlled trials usually refer to dummy medications (e.g., sugar pills) which in theory are indistinguishable from the treatment under investigation and which serve as methodological tools to screen out confounders that are associated with clinical trials (e.g., spontaneous remission, regression to the mean) [2]. In contrast, placebos in clinical practice are interventions without known specific effects on the treated condition but used with the goal of achieving positive outcomes by eliciting placebo effects (e.g., through treatment expectations) [3].”

Methods:

- I know this is a nested study, but this qualitative study needs to stand alone re information needed by the reader. We do not want to have to have the trial design open to understand the present study fully. In the study participant section, please add how they were recruited to the trial, adverts? Were they students? Etc.
- o We thank the reviewer for this comment. The following information was added in the method section: “Thirty participants from the intervention groups (i.e., OPR-, OPR+, and DP) out of the total sample of 160 were randomly selected to take part in the nested qualitative study. Participants were healthy adults from the general population recruited through advertisements.”

Discussion:

- In the discussion, authors state the below on line 463: “Whereas expectancy is linked to cognitive states and a prediction of the future, hope is not so much a prediction but rather more an existential stance [36]”. It would be useful if they could go further and unpack what they mean by existential stance, do they mean a trait? Further explication here would be helpful for readers.

o We thank the reviewer for pointing this out. We decided to put it simpler. The sentence now reads: “Whereas expectancy is linked to cognitive states and a prediction of the future, hope is not so much a prediction but rather more an emotional state”

- I think it would be useful to discuss the implications of this work further. Although implications are discussed in relation to the rationale provided for open label placebos, and this appears to be the key focus, I think the authors, albeit briefly need to look beyond that. Is the aim of this research to make open label placebos a part of clinical practice? If so, I think there should be some consideration given for what the findings of this study might mean for application in clinical practice. For instance, I think it is important that most seem to express scepticism.

o We thank the reviewer for this input. We believe that implications for the clinical practice should be discussed with caution and that only preliminary indications can be outlined. Therefore, the following statement was now added to the conclusion section: “Considering that this qualitative analysis stems from an experimental trial, clinical implications should be outlined with caution: Our findings provide preliminary indications that the OLP approach might only be suitable for a subgroup of patients who are not too skeptical about the efficacy of the OLP therapy and who are ready “to just give it a try” – maybe also because they feel that OLPs are their last treatment option.”

Reviewer: 2 (Miss Zhen Zhou, University of Tasmania Menzies Institute for Medical Research, First Affiliated Hospital of Anhui Medical University)

- Thank you for inviting me to review this work. This is a very nicely written paper with important information conveyed that will provide many implications for the improvement of OLP rationales. The study methods and findings were well presented and the authors comprehensively and meticulously discussed the study findings. I only have few minor suggestions for the authors.

o We thank Miss Zhen Zhou for her positive feedback.

- First, the authors may consider clearly stating the rationales that were given to the OPR+ group in the OLP trial where their interviewees had been recruited (the first three rationales mentioned in their introduction). This may help readers to understand how these informed rationales may impact the OPR+ group’s responses to the questions related to the discussion points. Please ignore this suggestion if the authors already did that.

o We thank the reviewer for pointing this out. The rationales have now been translated from German into English and can now be found in the Supplement (eAppendix 1).

- I wonder it may be worth mentioning that the OLP trial in which their study was nested had not included Psychology or Medicine students because they potentially have a previous knowledge of placebo mechanisms and effects?

o We agree that it is worth to be more specific about this point. Therefore, the following statement was added in the method section: “None of the participants were Psychology or Medicine students due to potential prior knowledge regarding placebo mechanisms and effects.”

- It would be useful if the authors can add the concrete number of participants after some terms they used such as ‘the vast majority of interviewees’, ‘The majority of?’ I know they had this information included in the appendix, but it often takes time for readers to find this information in the supplementary documents.

o We thank the reviewer for this input. However, we deliberately decided to not include the number of statements in the manuscript. The reason for this is as follows: The absolute number of statements (i.e., not the number participants who emphasized a statement) is listed in the supplement. Since – in line with the thematic approach – we counted every statement, the reference number is varying. We feel like the changing reference number (that is not in line with the total amount of participants) might be confusing for the reader. We hope that our reason to not list the number of statements in the manuscript is understandable for the reviewer.

- In the first paragraph of the discussion section, the authors stated that their study compared participants’ attitudes toward OLP with the four discussion points. Shouldn’t it be ‘three’ discussion

points as they claimed that they were unable to explore the fourth point due to the experimental design of their OLP trial?

o We corrected the wording as suggested and it now reads: “[...] to compare participants’ attitudes towards (open-label) placebos with the three discussion points of the conventional OLP disclosures [...].”

o We also changed other passages where we referred to “four” instead of “three” discussion points.

• Also in this paragraph, the authors mentioned the results of the OLP trial, which are not the findings of their qualitative study. They may consider moving this part to the method section after the brief introduction of the trial.

o We thank the reviewer for this suggestion. We moved the description of the quantitative findings to the method section and slightly changed the wording: “In the quantitative part that has been published elsewhere [30], we found that the OPR+ group exhibited a significant reduction of subjective heat pain ratings that did not differ from the DP group.”

VERSION 2 – REVIEW

REVIEWER	Adam Geraghty University of Southampton, Primary Care and Population Sciences
REVIEW RETURNED	29-Jul-2021

GENERAL COMMENTS	I thank the authors for addressing all my comments. Their response was very clear. I think this paper makes a useful and important contribution.
--

REVIEWER	Zhen Zhou University of Tasmania Menzies Institute for Medical Research
REVIEW RETURNED	16-Jul-2021

GENERAL COMMENTS	All my comments have been well-addressed by the authors, and the revised manuscript reads very well!
--